# Jointly Canonicalizing and Linking Open Knowledge Base via Unified Embedding Learning

## ABSTRACT

Recent years have witnessed increasing attention on the semantic knowledge integration between curated knowledge bases (CKBs) and open knowledge bases (OKBs), which is non-trivial due to the intrinsically heterogeneous features involved in CKBs and OKBs. OKB canonicalization (i.e., grouping synonymous noun phrases and relation phrases) and OKB linking (i.e., linking noun phrases and relation phrases with their corresponding entities and relations in a CKB) are regarded as two vital tasks to achieve the knowledge integration. Although these two tasks are inherently complementary with each other, previous studies just solve them separately or via superficial interaction. To address this issue, we propose CLUE, a novel framework that jointly encodes the OKB and CKB into a unified embedding space, to tackle OKB canonicalization and OKB linking simultaneously and make them benefit each other reciprocally. We design an expectation-maximization (EM) based approach to iteratively refine the unified embedding space via performing seed generation and embedding refinement alternately, by leveraging the deep interaction between OKB canonicalization and OKB linking. Curriculum learning is employed to yield high-quality canonicalization seeds and linking seeds adaptively, according to two elaborately designed metrics (i.e., a margin-based linking metric and an entropy-based cluster metric). A thorough experimental study over two public benchmark data sets demonstrates that our proposed framework CLUE consistently outperforms state-of-the-art baselines for the task of OKB canonicalization (resp. OKB linking) in terms of average F1 (resp. accuracy).

## KEYWORDS

Open Knowledge Base Canonicalization, Open Knowledge Base Linking, Unified Embedding Learning

## 1 INTRODUCTION

Knowledge bases (KBs), which store factual knowledge about real-world entities, have experienced a strong momentum for motivating various knowledge-driven applications such as question answering [18, 28] and recommendation system [49]. Generally, KBs could be classified into two categories: **1)** curated knowledge bases (CKBs) [12, 57], like notable Wikidata [46], DBpedia [1] and YAGO [38], are usually constructed via crowdsourcing [52] and closed information extraction [6, 26] techniques, requiring manually pre-specified ontology and significant human involvement; **2)** open knowledge bases (OKBs) such as ReVerb [11], OPIEC [16] and DefIE [4], are composed of open-domain triples extracted from massive unstructured text by open information extraction [10, 54], which operates at Web scale and is efficient and highly adaptable for its unsupervised manner. In general, the coverage and diversity of OKBs are much higher than CKBs. Accordingly, it is synergistic to integrate factual triples from OKBs into existing CKBs for enriching them.

Nevertheless, due to their different construction strategies, CKBs and OKBs contain intrinsically heterogeneous features, which are obvious obstacles for semantic knowledge integration between them. To be specific, each entity in the CKB is canonicalized and well defined with a unique identifier. In contrast, an entity in the OKB is not represented by a unique identifier but referred to by multiple distinct noun phrases (NPs), which is called the entity name variation problem for the OKB. For instance, there are two triples in an OKB, i.e., *<Elon Musk, is the CEO of, Tesla>* and *<Elon Reeve Musk, is the Chief Executive Officer of, SpaceX>*. It can be seen that *Elon Musk* and *Elon Reeve Musk* are two distinct noun phrases from different OKB triples, but referring to the same unique entity **"Elon Musk"** in a CKB, which is unaware for machines unfortunately. When we use the term *Elon Musk* as the query to retrieve the factual OKB triples for enriching the CKB about the entity **"Elon Musk"**, we cannot obtain the factual triples associated with the noun phrase *Elon Reeve Musk* from the OKB (e.g., the second OKB triple shown above), which leads to the problem of inadequate knowledge integration. Armed with this insight, OKB canonicalization [7, 14, 23, 37, 45] and OKB linking [27] are proposed as two essential tasks for integrating knowledge of OKBs and CKBs.

- **OKB canonicalization** is the task of canonicalizing OKB triples, by clustering noun (or relation) phrases with the same semantic meaning into a group. With regard to the two aforementioned OKB triples, OKB canonicalization task aims at clustering the two synonymous noun phrases *Elon Musk* and *Elon Reeve Musk* into one group and clustering the two synonymous relation phrases (i.e., *is the CEO of* and *is the Chief Executive Officer of*) into another group.

- **OKB linking** is the task of linking noun (resp. relation) phrases in an OKB with their corresponding entities (resp. relations) in a CKB. For the aforementioned example, OKB linking task needs to link both noun phrases *Elon Musk* and *Elon Reeve Musk* with their corresponding entity **"Elon Musk"** in the CKB.

The two task definitions above show that OKB canonicalization and OKB linking are tightly related and inherently complementary with each other [27]. Thus, it will be beneficial to solve these two tasks together in a unified framework to make them reinforce each other. JOCL [27] is the first work to handle OKB canonicalization and OKB linking tasks jointly and achieves promising results. However, JOCL firstly models these two tasks separately based on the factor graph model, and then adds consistency signals into the factor graph to mutually constrain the two tasks at the output level, omitting the deep interaction between them. Additionally, JOCL relies on some hand-crafted features, which make the approach labor-intensive and also limited by the availability of extra third-party resources such as PPDB [34] and Stanford Knowledge Base Population (KBP) system [42].

In addition, once we neglect the entity name variation problem existing in the OKB, which is an intrinsic difference between CKBs and OKBs, OKB linking task could be considered similar to the well-studied KB entity alignment problem [22, 41] for aligning NPs in

the OKB with their corresponding entities in the CKB. A multitude of KB entity alignment studies [30, 40, 53, 55] have proposed sophisticated solutions to handle KB entity alignment as a combinatorial optimization problem based on the 1-to-1 mapping assumption [22]: each entity in one KB has one and only one counterpart in the other KB. However, in fact, this assumption does not hold true for the OKB linking task as there are usually multiple NPs in the OKB referring to the same entity in the CKB, which inevitably leads to the incapability of these solutions to resolve the OKB linking task. Still, it is desirable to resort to some other KB entity alignment approaches [56, 58] following a simple greedy search strategy to output equivalences between NPs and entities to address the OKB linking task. Nevertheless, since these approaches have implicitly assumed that the to-be-aligned KBs are canonicalized, they unsurprisingly fall short of leveraging reciprocal benefits from the OKB canonicalization task and consequently perform unsatisfactorily over the OKB linking task, as we show in the experiments.

To address the above issues, we propose **CLUE**, a novel framework for jointly **C**anonicalizing and **L**inking OKBs via **U**nified **E**mbedding learning. Based on the learned universal embedding space, which allows OKB canonicalization and OKB linking to share latent features and naturally reinforce each other, CLUE could resolve these two tasks simultaneously with superior performance. Firstly, a multi-task unified embedding learning model is proposed to jointly encode the OKB and CKB into a shared universal embedding space by taking advantage of correlated information from three highly related tasks. Subsequently, we propose an expectation-maximization (EM) based approach to further refine the universal embedding space in an iterative manner, by leveraging the deep interaction of OKB canonicalization and OKB linking based on the following two assumptions:

**Assumption 1:** *If two noun phrases are linked to the same entity in high confidence via OKB linking, then these two noun phrases could be regarded as a canonicalization seed for OKB canonicalization.*

**Assumption 2:** *If two noun phrases are clustered to the same group in high quality via OKB canonicalization and meanwhile one of them is linked to an entity in high confidence via OKB linking, then the other noun phrase and the referred entity could be regarded as a linking seed for OKB linking.*

Specifically, in the expectation step, we exploit curriculum learning to derive high-quality canonicalization seeds and linking seeds in an adaptive manner, according to a couple of well-designed metrics: (1) a margin-based linking metric to assess the confidence of linking pairs; and (2) an entropy-based cluster metric to evaluate the quality of canonicalization clusters. Next, in the maximization step, the canonicalization seeds and linking seeds derived in the expectation step are leveraged by the multi-task unified embedding learning model to further refine the universal embedding space.

Our major contributions can be summarized as follows:

- To our best knowledge, we are the first to jointly encode the OKB and CKB into a shared and unified embedding space, to tackle OKB canonicalization and OKB linking simultaneously and make them reinforce each other.
- We propose an EM based approach to iteratively promote the universal embedding space via performing seed generation and

embedding refinement alternately, by exploiting the deep interaction between OKB canonicalization and OKB linking.

- In the expectation step, we employ curriculum learning to derive high-quality canonicalization seeds and linking seeds adaptively, according to two elaborately designed metrics (i.e., a margin-based linking metric and an entropy-based cluster metric).
- The experimental results on two public benchmark data sets demonstrate that the proposed CLUE significantly outperforms all the baseline methods for both OKB canonicalization and OKB linking tasks.

## 2 NOTATIONS AND PROBLEM DEFINITION

In this section, we introduce some important notations and define the task of joint OKB canonicalization and linking.

A CKB $\mathcal{K}_C$ is formed with $\mathcal{E}$, the set of entities, and $\mathcal{R}$, the set of relations. The sets of noun phrases (NPs) and relation phrases (RPs) in an OKB $\mathcal{K}_O$ are denoted by $\mathcal{N}$ and $\mathcal{P}$ respectively. It is worth noting that the sets $\mathcal{E}$ and $\mathcal{N}$ (as well as $\mathcal{R}$ and $\mathcal{P}$) are disjoint. Both CKB and OKB consist of triples, and we use $T_c^+$ and $T_o^+$ to denote their corresponding sets of triples. For triples $(h_c, r_c, t_c) \in T_c^+$ and $(h_o, r_o, t_o) \in T_o^+$, we have $h_c, t_c \in \mathcal{E}$, $h_o, t_o \in \mathcal{N}$, $r_c \in \mathcal{R}$ and $r_o \in \mathcal{P}$.

PROBLEM 1 (**JOINT OKB CANONICALIZATION AND LINKING**). *Given an OKB and a CKB, the goal is to simultaneously (i) cluster synonymous NPs (or RPs) in the OKB into a group (i.e., **OKB canonicalization**) and (ii) link each NP (resp. RP) in the OKB with its corresponding entity (resp. relation) in the CKB (i.e., **OKB linking**).*

Our framework works in a universal embedding space, denoted by $\Theta$, in which each element of a triple $(h, r, t)$ is represented by an embedding vector in the form of $(\mathbf{h}, \mathbf{r}, \mathbf{t})$. In the rest of this paper, the bold-faced letter represents the embedding vector of the corresponding element by convention.

## 3 THE PROPOSED FRAMEWORK: CLUE

### 3.1 Framework Overview

We provide an overview of our proposed framework CLUE in Figure 1. First of all, a multi-task unified embedding learning model (Section 3.2) is proposed to jointly encode the OKB and CKB into a shared and unified embedding space $\Theta_0$ (❶ in Figure 1) by exploiting correlated information from three highly related tasks.

After that, to further improve the performance of OKB canonicalization and OKB linking, we propose to leverage the deep coupling of these two tasks to iteratively refine the universal embedding space based on the expectation-maximization (EM) [9] algorithm. The EM algorithm is an iterative statistical technique to optimize model parameters with incomplete data. In our scenario, the EM process alternates between performing an E-step (❽ in Figure 1) (Section 3.3), which automatically discovers potential training seeds (including canonicalization seeds and linking seeds) with the current model parameters (i.e., the unified embedding space $\Theta_{k-1}$ learned via the previous iteration $k - 1$), and an M-step (❾ in Figure 1) (Section 3.4), which optimizes the model parameters (i.e., embedding refinement) by leveraging the training seeds generated in the E-step to obtain the refined embedding space $\Theta_k$ of the current iteration $k$. Subsequently, the next E-step could be executed according to the refined embedding space $\Theta_k$ output by the previous M-step to derive more high-quality training seeds, and so on. It

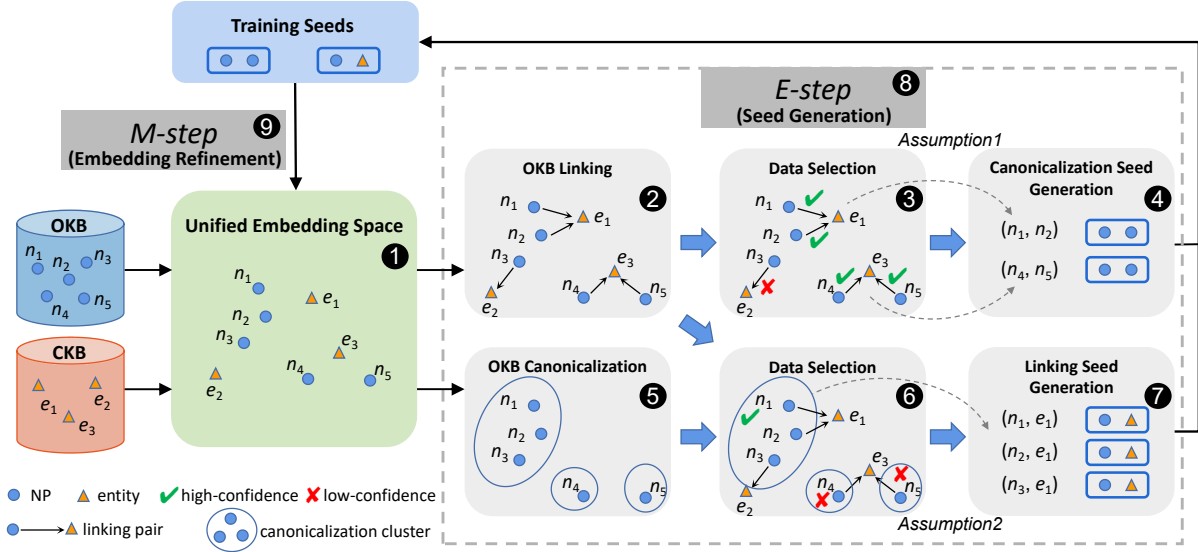

Figure 1: The overall architecture of framework CLUE.

can be seen that this iterative procedure performs seed generation and embedding refinement alternately to enhance the quality of the universal embedding space. When reaching the pre-defined number of iterations $K$, OKB canonicalization and OKB linking could be performed based on the finally refined embedding space $\Theta_K$ to output the final results of these two tasks.

## 3.2 Multi-task Unified Embedding Learning

Multi-task learning [5] is a classical learning paradigm in machine learning, which aims to learn knowledge from several different tasks simultaneously by calculating the loss function with respect to multiple tasks. The hypothesis behind multi-task learning is that these tasks are closely related, and thus, training with the same model could share knowledge and benefit each other. Since the two tasks of OKB canonicalization and OKB linking are inherently complementary with each other [27], we could naturally formulate them along with an auxiliary task of structure learning as a multi-task learning problem and take advantage of correlated information from each task to learn a shared universal embedding space $\Theta_0$ for the OKB and CKB. Accordingly, we define the overall loss function for the unified embedding learning according to these three parts:

$$\mathcal{L} = \mathcal{L}_{stru} + \mathcal{L}_{cano} + \mathcal{L}_{link} \qquad (1)$$

where $\mathcal{L}_{stru}$, $\mathcal{L}_{cano}$ and $\mathcal{L}_{link}$ denote the loss function of structure learning, OKB canonicalization learning and OKB linking learning respectively. In the following, we will introduce these three parts in details.

*3.2.1 Structure Learning.* For both OKB and CKB, the structural information is preserved in the form of relational triples, so it is reasonable to learn the structure embedding for OKB and CKB in the same way via a KB embedding model [19, 47]. In this paper, we choose the widely-used and simple KB embedding model TransE [3] to give prominence to the effects of our framework. Note that any other KB embedding model is also applicable, but is not the focus of this paper and left for future exploration. In TransE, relations (resp. RPs) are considered as translations from head entities (resp.

subject NPs) to tail entities (resp. object NPs). We could learn the structure embedding for a KB $\mathcal{K}$ by minimizing a margin-based loss as follows:

$$\mathcal{L}_{stru}^{\mathcal{K}} = \sum_{(h,r,t)\in T^+} \sum_{(h',r',t')\in T^-} \left[ \gamma_1 + \|\mathbf{h} + \mathbf{r} - \mathbf{t}\| - \|\mathbf{h'} + \mathbf{r'} - \mathbf{t'}\| \right]_+ \qquad (2)$$

where $[\cdot]_+ = \max(0, \cdot)$, $\|\cdot\|$ is the $L_1$ vector norm, and $\gamma_1 > 0$ is a margin hyperparameter. $T^+$ denotes the set of all relational triples in a KB $\mathcal{K}$, and $T^-$ denotes the set of negative relational triples, which are generated by replacing the head entity (resp. subject NP) or tail entity (resp. object NP) of a valid triple in $T^+$ with a random entity (resp. NP) in $\mathcal{E}$ (resp. $\mathcal{N}$). It is worth mentioning that this loss is naturally applicable to the structure learning for both OKB and CKB with relational triples $T_o^+$ and $T_c^+$ respectively. Combining the structure loss $\mathcal{L}_{stru}^{\mathcal{K}_O}$ for the OKB and $\mathcal{L}_{stru}^{\mathcal{K}_C}$ for the CKB, the joint loss of structure learning is given as below:

$$\mathcal{L}_{stru} = \mathcal{L}_{stru}^{\mathcal{K}_O} + \mathcal{L}_{stru}^{\mathcal{K}_C} \qquad (3)$$

*3.2.2 OKB Canonicalization Learning.* Given a set of canonicalization seeds $\mathcal{V}_0^+ = \{(n_i, n_j)\}$, where $n_i \in \mathcal{N}$ and $n_j \in \mathcal{N}$ are synonymous NPs, the goal of OKB canonicalization learning is to narrow the embedding distance between $n_i$ and $n_j$ such that they are more likely to be clustered into the same group in the task of OKB canonicalization. Following [25], we design a contrastive loss function to enforce the synonymous NPs (i.e., NPs in each canonicalization seed) to get close, while other nonsynonymous NPs (i.e., NPs generated by negative sampling) are pushed away from each other in the universal embedding space, formulated as follows:

$$\mathcal{L}_{cano} = \sum_{(n_i, n_j)\in\mathcal{V}_0^+} \|\mathbf{n}_i - \mathbf{n}_j\| + \alpha_1 \sum_{(n_{i'}, n_{j'})\in\mathcal{V}_0^-} \left[ \gamma_2 - \|\mathbf{n}_{i'} - \mathbf{n}_{j'}\| \right]_+ \qquad (4)$$

where $\alpha_1$ is a balance factor, $\gamma_2 > 0$ is a margin hyperparameter and $\mathcal{V}_0^-$ denotes the set of negative canonicalization seeds, which are generated by replacing either $n_i$ or $n_j$ in a positive seed $(n_i, n_j)$ with a random NP in $\mathcal{N}$.

*3.2.3 OKB Linking Learning.* Given a set of linking seeds $\mathcal{S}_0^+ = \{(n_i, e_j)\}$, where an NP $n_i \in \mathcal{N}$ is linked to an entity $e_j \in \mathcal{E}$, the goal of OKB linking learning is to shorten the embedding distance between $n_i$ and $e_j$ such that $n_i$ is more likely to be linked to $e_j$ in the task of OKB linking. Similar to OKB canonicalization learning introduced in Section 3.2.2, we devise a loss function based on contrastive learning to enforce the NP and its corresponding entity in each linking seed to get close, while other entities are pushed away from this NP in the universal embedding space, which is formulated as follows:

$$\mathcal{L}_{link} = \sum_{(n_i, e_j) \in \mathcal{S}_0^+} \| \mathbf{n}_i - \mathbf{e}_j \| + \alpha_2 \sum_{(n_{i'}, e_{j'}) \in \mathcal{S}_0^-} \left[ \gamma_3 - \| \mathbf{n}_{i'} - \mathbf{e}_{j'} \| \right]_+ \tag{5}$$

where $\alpha_2$ is a balance factor, $\gamma_3 > 0$ is a margin hyperparameter and $\mathcal{S}_0^-$ denotes the set of negative linking seeds, which are generated by replacing $e_j$ in a positive seed $(n_i, e_j)$ with a random entity in $\mathcal{E}$. Note that we do not generate a negative linking seed by randomly replacing $n_i$ because OKB linking is an N-to-1 mapping problem.

## 3.3 Expectation-step: Seed Generation

Having obtained the current unified embedding space, we could generate more training seeds automatically by leveraging the coupling of OKB canonicalization and OKB linking according to the two assumptions introduced in Section 1. To be specific, in the beginning of E-step at iteration $k$, OKB canonicalization and OKB linking are conducted based on the universal embedding space $\Theta_{k-1}$ learned via the previous iteration $k-1$. Subsequently, new canonicalization seeds could be yielded with reciprocal benefits from the OKB linking task by incorporating the knowledge of high-confidence linking pairs, and vice versa. We elaborate the generation of canonicalization seeds and linking seeds in the following.

*3.3.1 Linking-guided Canonicalization Seed Generation.* As introduced in **Assumption 1**, it is helpful to exploit the result of OKB linking to guide OKB canonicalization in the next iteration by providing new canonicalization seeds. We first depict the process of OKB linking to get the result of linking pairs (❷ in Figure 1). Then, a margin-based linking metric is proposed to distinguish high-confidence ones from all linking pairs (❸ in Figure 1). Finally, new canonicalization seeds are generated using these high-confidence linking pairs according to **Assumption 1** (❹ in Figure 1).

**OKB Linking.** Given an NP $n \in \mathcal{N}$, we employ a vector similarity metric $f(\cdot)$ to calculate the embedding similarity score between $n$ and each of its candidate entities in the universal embedding space. Then, we could choose the candidate entity with the highest similarity score as the predicted corresponding entity $e^*(n)$ for the NP $n$ based on the following formula:

$$e^*(n) = \arg\max_{e_i \in E(n)} f(\mathbf{n}, \mathbf{e}_i) \tag{6}$$

Here, $E(n) \subseteq \mathcal{E}$ is the set of candidate entities for $n$ and $f(\cdot)$ is instantiated as cosine similarity in our experiments. Consequently, the NP $n$ and its predicted corresponding entity $e^*(n)$ form a linking pair. The linking process for RPs is similar and omitted for brevity.

**Margin-based Linking Metric.** Since there are inevitable errors in the result of OKB linking and not every NP is linked to its gold mapping entity correctly, it is crucial to identify high-confidence

linking pairs to guarantee the quality of generated seeds. For this purpose, we propose a margin-based linking metric to assess the confidence of the linking pair, with a focus on how much the similarity score of the predicted corresponding entity deviates from the scores of other candidate entities, outlined as follows:

$$conf(n, e^*(n)) = \text{sigmoid}\left( \frac{f(n, e^*(n)) - f(n, e^{**}(n))}{f(n, e^*(n))} \right) \tag{7}$$

where $f(n, e^*(n))$ and $f(n, e^{**}(n))$ are the highest and second highest similarity scores of candidate entities with respect to the NP $n$ respectively. A large margin between them represents a high confidence of the linking pair $(n, e^*(n))$.

**Canonicalization Seed Generation.** If there are high-confidence linking pairs $(n_i, e_m)$ and $(n_j, e_m)$ in the result of OKB linking, which indicates that both $n_i$ and $n_j$ are linked to $e_m$ in high confidence, $n_i$ and $n_j$ can form a new canonicalization seed as $(n_i, n_j)$ according to **Assumption 1** (❹ in Figure 1). At iteration $k$, the set of newly generated canonicalization seeds $\mathcal{V}_k^{new}$ is added into the original canonicalization seed set $\mathcal{V}_0^+$ to get the set of updated training canonicalization seeds $\mathcal{V}_k^+$ as follows:

$$\mathcal{V}_k^+ = \mathcal{V}_0^+ \cup \mathcal{V}_k^{new} \tag{8}$$

Our strategy for the selection of high-confidence linking pairs would be introduced in Section 3.3.3.

*3.3.2 Canonicalization-guided Linking Seed Generation.* Based on **Assumption 2**, the result of OKB canonicalization is beneficial for OKB linking in the next iteration via offering instructive information embedded in the new linking seeds. Similar to Section 3.3.1, we first introduce the process of OKB canonicalization to obtain the result of canonicalization clusters (❺ in Figure 1). Then, an entropy-based cluster metric is proposed to recognize high-quality ones from all canonicalization clusters (❻ in Figure 1). Ultimately, new linking seeds are yielded from these high-quality canonicalization clusters according to **Assumption 2** (❼ in Figure 1).

**OKB Canonicalization.** Based on the universal embedding space, we cluster NPs and RPs into groups by performing hierarchical agglomerative clustering (HAC) according to the cosine distance.

**Entropy-based Cluster Metric.** The result obtained from OKB canonicalization should not be fully trusted since NPs cannot always be clustered accurately. Hence, only high-quality canonicalization clusters should be used for linking seed generation to avoid error propagation during the iterative process. Armed with this insight, we propose an entropy-based cluster metric to evaluate the quality of canonicalization clusters, following the intuition that the cluster with lower uncertainty is of higher quality. For a non-singleton canonicalization cluster $w$, we quantify its uncertainty $H(w)$ by measuring the disagreement among the linking predictions of NPs within $w$, which could be calculated in the form of Shannon entropy, a well-known uncertainty measurement theory. Since a high uncertainty indicates a low quality for the canonicalization cluster, we define the quality score for the canonicalization cluster $w$ as follows:

$$qual(w) = \exp(-H(w)) \tag{9}$$

$$H(w) = - \sum_{e_i \in A(w)} p(e_i) \cdot \log p(e_i) \tag{10}$$

$A(w)$ is the set of predicted corresponding entities for NPs in cluster $w$, defined as: $A(w) = \{e^*(n)|n \in w\}$, and $p(e_i)$ is the probability of a randomly chosen $n \in w$ being linked to entity $e_i$, taking the linking confidence (defined in Eq. 7) into consideration as:

$$p(e_i) = \frac{\sum_{n \in w} conf(n, e^*(n)) \cdot \mathbb{1}\left[e^*(n) = e_i\right]}{\sum_{n \in w} conf(n, e^*(n))} \quad (11)$$

where $\mathbb{1}[x]$ is an indicator function whose value is 1 if condition $x$ is true, otherwise the value is 0.

**Linking Seed Generation.** For a high-quality canonicalization cluster $w$, the entity $e_j$ with the highest probability $p(e_j)$ is regarded as the anchor entity of $w$. According to **Assumption 2**, the entity $e_j$ can form a new linking seed with each NP $n_i \in w$ as $(n_i, e_j)$ (❼ in Figure 1). At iteration $k$, the set of newly yielded linking seeds $\mathcal{S}_k^{new}$ is added into the original linking seed set $\mathcal{S}_0^+$ to obtain the set of updated training linking seeds $\mathcal{S}_k^+$ as follows:

$$\mathcal{S}_k^+ = \mathcal{S}_0^+ \cup \mathcal{S}_k^{new} \quad (12)$$

Our strategy for the selection of high-quality canonicalization clusters is introduced in the following.

*3.3.3 Curriculum Learning based Data Selection.* According to the aforementioned metrics (Eq. 7 and Eq. 9), we could obtain the confidence score $conf(n, e^*(n))$ for each linking pair $(n, e^*(n))$ and the quality score $qual(w)$ for each canonicalization cluster $w$. Subsequently, it is necessary to establish a criterion for these scores, based on which we could identify high-confidence linking pairs and high-quality canonicalization clusters. A fixed threshold is a naive approach but setting an appropriate threshold is challenging. It requires finding a balance between a too-small and a too-large threshold, both of which can negatively affect the performance. Besides, during the iterative EM process, the utilization of a fixed threshold is not flexible enough, failing to effectively incorporating feedback from the dynamic model parameters.

In order to address these issues, we design a curriculum learning [48] based data selection strategy to adaptively recognize high-confidence linking pairs and high-quality canonicalization clusters with the iteration progressing. Since the selection processes for high-confidence linking pairs and high-quality canonicalization clusters are similar, we will focus on illustrating the latter. Let $W$ represent the set of all canonicalization clusters, and we introduce a parameter $\boldsymbol{v} = [v_1, v_2, ..., v_{|W|}]$ to assign weights to each cluster, indicating whether it should be selected and how important it is as a high-quality canonicalization cluster. Our goal is to obtain the optimal weights $\boldsymbol{v^*} = [v_1^*, v_2^*, ..., v_{|W|}^*]$ in each iteration. Formally, the objective of our curriculum learning at iteration $k$ is defined as:

$$\underset{\boldsymbol{v} \in [0,1]^{|W|}}{\arg\min} \ J(\boldsymbol{v}; k) = \sum_{i=1}^{|W|} v_i l_s(qual(w_i)) + g(\boldsymbol{v}; k) \quad (13)$$

where $qual(w_i)$ is the quality score of the canonicalization cluster $w_i$ as defined in Eq. 9, and the function $l_s(x) = ln(2 - x)$ is used to make a negative non-linear mapping for $qual(w_i)$. Inspired by [20], we define the self-paced function $g(\boldsymbol{v}; k)$ to control the learning pace according to the iteration number $k$ as follows:

$$g(\boldsymbol{v}; k) = \exp(-\frac{1}{k}) \sum_{i=1}^{|W|} (\frac{1}{2}v_i^2 - v_i) \quad (14)$$

By substituting Eq. 14 into Eq. 13, we could get the closed-form optimal solution for $\boldsymbol{v^*} = [v_1^*, v_2^*, ..., v_{|W|}^*]$:

$$v_i^* = \begin{cases} 1 - l_s(qual(w_i))/\exp(-\frac{1}{k}) & l_s(qual(w_i)) < \exp(-\frac{1}{k}) \\ 0 & l_s(qual(w_i)) \geq \exp(-\frac{1}{k}) \end{cases} \quad (15)$$

where the optimal weight $v_i^* \in [0, 1]$ evaluates the importance of cluster $w_i$. When $v_i^* = 0$, it implies that the cluster $w_i$ should not be selected as a high-quality canonicalization cluster. In the initial stage, with a low value of $\exp(-\frac{1}{k})$, only canonicalization clusters with the highest quality scores would be selected as high-quality clusters. As the iteration progresses, the value of $\exp(-\frac{1}{k})$ would gradually increase, allowing for the inclusion of more canonicalization clusters with sub-optimal scores. By adaptively adjusting the selection criteria, this curriculum learning strategy enables a balanced learning pace to select the most suitable high-quality clusters in each iteration, thereby enhancing the learning process and improving overall performance.

**Reliability Score.** Despite the well-designed selection process, the high-confidence linking pairs and high-quality canonicalization clusters are not guaranteed to be absolutely correct, which may accordingly lead to errors in the newly generated canonicalization seeds and linking seeds. To alleviate this, we propose assigning a reliability score to each generated seed based on the optimal weights $\boldsymbol{v^*}$ of the selected high-confidence linking pairs and high-quality canonicalization clusters.

For high-confidence linking pairs $(n_i, e_m)$ and $(n_j, e_m)$, we define the reliability score $R_{cano}(n_i, n_j)$ for the generated canonicalization seed $(n_i, n_j)$ as:

$$R_{cano}(n_i, n_j) = v_i^* \cdot v_j^* \quad (16)$$

where $v_i^*$ and $v_j^*$ denote the optimal weights of linking pairs $(n_i, e_m)$ and $(n_j, e_m)$, respectively. Similarly, for a high-quality canonicalization cluster $w$ with its anchor entity $e_j$ and each NP $n_i \in w$, we define the reliability score $R_{link}(n_i, e_j)$ for the generated linking seed $(n_i, e_j)$ as:

$$R_{link}(n_i, e_j) = v^* \quad (17)$$

where $v^*$ is the optimal weight of the canonicalization cluster $w$.

## 3.4 Maximization-step: Embedding Refinement

In the M-step, based on the new training seeds generated in the E-step, we could further refine the universal embedding space. As shown in Figure 1 (❷), the NP $n_3$ is linked to the entity $e_2$ based on the current universal embedding space $\Theta_{k-1}$. Nevertheless, $e_1$ is the correct corresponding entity for the NP $n_3$ in fact. By exploiting the newly generated linking seed $(n_3, e_1)$ shown in Figure 1 (❼), the embedding distance between $n_3$ and $e_1$ would be shortened in the refined embedding space $\Theta_k$ in the M-step so that $n_3$ is more likely to be linked to $e_1$ in the next iteration, which would improve the performance of OKB linking. Similarly, $n_4$ and $n_5$ are synonymous NPs in reality which should be clustered into the same group in OKB canonicalization, but they are wrongly clustered into different groups based on the current universal embedding space $\Theta_{k-1}$ shown in Figure 1 (❺). By utilizing the newly generated canonicalization seed $(n_4, n_5)$ shown in Figure 1 (❹), the embedding distance between $n_4$ and $n_5$ would be reduced in the refined embedding space $\Theta_k$ in the M-step so that $n_4$ and $n_5$ are more likely to be clustered into the same group in the next iteration, which would

**Table 1: Performance on OKB NP canonicalization task.**

| Method | ReVerb45K | | | | OPIEC59K | | | |
|---|---|---|---|---|---|---|---|---|
| | Macro F1 | Micro F1 | Pairwise F1 | Average F1 | Macro F1 | Micro F1 | Pairwise F1 | Average F1 |
| Morph Norm [11] | 0.281 | 0.699 | 0.653 | 0.544 | 0.476 | 0.222 | 0.186 | 0.294 |
| Text Similarity [14] | 0.543 | 0.821 | 0.689 | 0.684 | 0.480 | 0.228 | 0.192 | 0.300 |
| IDF Token Overlap [14] | 0.598 | 0.571 | 0.505 | 0.558 | 0.457 | 0.225 | 0.190 | 0.290 |
| Attribute Overlap [14] | 0.598 | 0.599 | 0.587 | 0.595 | 0.474 | 0.226 | 0.187 | 0.295 |
| CESI [45] | 0.618 | 0.845 | 0.819 | 0.761 | 0.328 | 0.807 | 0.667 | 0.600 |
| SIST [23] | 0.691 | 0.889 | 0.823 | 0.801 | / | / | / | / |
| CUVA [7] | 0.661 | 0.845 | 0.855 | 0.794 | 0.128 | 0.789 | 0.686 | 0.534 |
| JOCL [27] | 0.684 | 0.892 | 0.877 | 0.818 | 0.337 | 0.907 | 0.922 | 0.722 |
| CMVC [37] | 0.662 | 0.881 | 0.893 | 0.812 | 0.521 | 0.909 | 0.878 | 0.769 |
| CLUE | 0.721 | 0.912 | 0.904 | **0.845** | 0.606 | 0.928 | 0.903 | **0.812** |

promote the performance of OKB canonicalization. It can be seen from the above examples that by exploiting the new training seeds generated in the E-step, the M-step could effectively refine the universal embedding space and achieve error correction, which is beneficial for both tasks of OKB canonicalization and OKB linking.

To be specific, the newly yielded canonicalization seeds and linking seeds with reliability scores could be applied to the multi-task unified embedding learning model described in Section 3.2 in an incremental manner. Since the knowledge from newly yielded seeds might be error-prone, a reliability score (Eq. 16 for canonicalization seeds and Eq. 17 for linking seeds) is accompanied with each new seed. Hence, we update the losses for OKB canonicalization learning and OKB linking learning at iteration $k$ in the M-step as:

$$\mathcal{L}'_{cano} = \sum_{(n_i, n_j) \in \mathcal{V}_k^+} R_{cano}(n_i, n_j) \left\| \mathbf{n}_i - \mathbf{n}_j \right\|$$
$$+ \alpha_1 \sum_{(n_{i'}, n_{j'}) \in \mathcal{V}_k^-} \left[ \gamma_2 - \left\| \mathbf{n}_{i'} - \mathbf{n}_{j'} \right\| \right]_+ \quad (18)$$

and

$$\mathcal{L}'_{link} = \sum_{(n_i, e_j) \in \mathcal{S}_k^+} R_{link}(n_i, e_j) \left\| \mathbf{n}_i - \mathbf{e}_j \right\|$$
$$+ \alpha_2 \sum_{(n_{i'}, e_{j'}) \in \mathcal{S}_k^-} \left[ \gamma_3 - \left\| \mathbf{n}_{i'} - \mathbf{e}_{j'} \right\| \right]_+ \quad (19)$$

where $\mathcal{V}_k^+$ and $\mathcal{S}_k^+$ are the sets of updated training canonicalization seeds and linking seeds at iteration $k$ obtained from Eq. 8 and Eq. 12 respectively, and $R_{cano}(n_i, n_j)$ and $R_{link}(n_i, e_j)$ are reliability scores calculated by Eq. 16 and Eq. 17 respectively. It is noted that the reliability score of each seed existing in both the original canonicalization seed set $\mathcal{V}_0^+$ and the original linking seed set $\mathcal{S}_0^+$ is set to 1 for simplicity. Finally, the overall loss function for learning the refined embedding space $\Theta_k$ in the M-step is defined as:

$$\mathcal{L} = \mathcal{L}_{stru} + \mathcal{L}'_{cano} + \mathcal{L}'_{link} \quad (20)$$

The iterative EM process is repeated until reaching the pre-defined number of iterations $K$, and the ultimately refined unified embedding space $\Theta_K$ can be used for performing OKB canonicalization and OKB linking to output the final results. The whole process of our framework CLUE is depicted in Algorithm 1 of Appendix A.

## 4 EXPERIMENTS

### 4.1 Experimental Settings

*4.1.1 Data Sets.* We perform experiments on two public and widely used benchmark OKB data sets: ReVerb45K [45] (45k triples, 15.5k

NPs and 22k RPs) and OPIEC59K [37] (59k triples, 22.8k NPs and 17k RPs), in which all NPs are annotated with their corresponding entities in Freebase [2] and Wikidata [46], respectively. To ensure a fair comparison, we adopt the same validation and test split as prior works, specifically JOCL [27] w.r.t. ReVerb45K and CMVC [37] w.r.t. OPIEC59K. It is worth noting that both data sets do not provide a designated training set, so we use the same canonicalization seeds collected by the previous work CMVC [37] as the original canonicalization seed set $\mathcal{V}_0^+$ to achieve a fair comparison. In addition, as CMVC does not address the OKB linking task, it does not collect linking seeds at all. To deal with this issue, we leverage the linking annotations in the validation set of both data sets as the original linking seed set $\mathcal{S}_0^+$, the same way as the previous work JOCL [27] for a fair comparison. We make the data sets and source code used in this paper publicly available for future research[1].

*4.1.2 Metrics.* For the task of OKB canonicalization, we utilize average F1 (i.e., averaging macro F1, micro F1 and pairwise F1) as the standard comprehensive evaluation metric, following previous OKB canonicalization studies [7, 14, 23, 27, 37, 45]. For the task of OKB linking, we adopt accuracy as the evaluation metric, the same as the previous study JOCL [27] and entity linking methods [36]. Detailed descriptions of these metrics are provided in Appendix B. More implementation details are introduced in Appendix C.

### 4.2 OKB Canonicalization Task

*4.2.1 OKB NP Canonicalization.* We compare the performance of our method CLUE in OKB NP canonicalization with several state-of-the-art approaches (i.e., Morph Norm [11], Text Similarity [14], IDF Token Overlap [14], Attribute Overlap [14], CESI [45], SIST [23], CUVA [7], JOCL [27] and CMVC [37]). A detailed introduction to these baselines is provided in Appendix D.1.

The experimental results of all the methods for OKB NP canonicalization are shown in Table 1. Specifically, all the baseline results on ReVerb45K are derived directly from JOCL [27] or their respective papers [7, 37]. The experimental results of baselines except JOCL on OPIEC59K are taken from CMVC [37] directly, and we execute the open-source JOCL over OPIEC59K to obtain its result. As shown in Table 1, our proposed CLUE significantly outperforms all competitive baselines in terms of average F1 on both data sets, indicating the superiority of CLUE for OKB NP canonicalization. Despite the fact that in comparison to our CLUE, the state-of-the-art baseline CMVC leverages a more context view based on the triple's source context, CLUE still promotes by 3.3 (resp. 4.3) percentages

---

[1]https://anonymous.4open.science/r/CLUE-5628/README.md

**Table 2: Performance on OKB RP canonicalization task.**

| Method | Macro F1 | Micro F1 | Pairwise F1 | Average F1 |
|---|---|---|---|---|
| *ReVerb45K* | | | | |
| AMIE [15] | 0.703 | 0.820 | 0.760 | 0.761 |
| PATTY [33] | 0.782 | 0.872 | 0.802 | 0.819 |
| SIST [23] | 0.875 | 0.872 | 0.845 | 0.864 |
| JOCL [27] | 0.848 | 0.923 | 0.851 | 0.874 |
| CMVC [37] | 0.853 | 0.928 | 0.856 | 0.879 |
| CLUE | 0.871 | 0.917 | 0.858 | **0.882** |
| *OPIEC59K* | | | | |
| AMIE [15] | 0.595 | 0.800 | 0.631 | 0.675 |
| CESI [45] | 0.699 | 0.752 | 0.628 | 0.693 |
| JOCL [27] | 0.722 | 0.804 | 0.636 | 0.721 |
| CMVC [37] | 0.542 | 0.854 | 0.770 | 0.722 |
| CLUE | 0.654 | 0.863 | 0.685 | **0.734** |

compared with CMVC in terms of average F1 over ReVerb45K (resp. OPIEC59K), which confirms that jointly encoding the OKB and CKB into a unified embedding space is a promising way to address the task of OKB NP canonicalization.

*4.2.2 OKB RP Canonicalization.* In addition to the aforementioned baselines CESI [45], SIST [23], JOCL [27] and CMVC [37], we add AMIE [15] and PATTY [33] as baselines on the task of OKB RP canonicalization, which are described in detail in Appendix D.2.

Since RPs are not annotated in ReVerb45K and OPIEC59K, we randomly sample 35 non-singleton RP clusters over each data set and manually label them as the ground truth for RP canonicalization, which is the same as previous studies [23, 27, 37]. The experimental results for OKB RP canonicalization are shown in Table 2. To be specific, we take the experimental results on ReVerb45K from JOCL [27] directly for all the baselines except CMVC, which is not included in [27] and we take its result from its own paper [37]. Apart from JOCL [27], which is evaluated via running its open-source solution, all the baseline results on OPIEC59K are obtained from CMVC [37] directly. It can be seen that CLUE surpasses all baselines with respect to average F1 on both data sets, which validates the effectiveness of CLUE for OKB RP canonicalization. Although the state-of-the-art CMVC performs well by incorporating prior knowledge derived from external resources in the form of RP seed pairs, CLUE does not rely on such information. In comparison to CMVC, CLUE improves by approximately 0.3 (resp. 1.2) percentages in terms of average F1 over ReVerb45K (resp. OPIEC59K), indicating its superiority.

## 4.3 OKB Linking Task

*4.3.1 OKB Entity Linking.* Besides the aforementioned JOCL [27] which resolves both OKB canonicalization and OKB linking, we additionally select several off-the-shelf entity linking tools (i.e., Tagme [13], Spotlight [31], Falcon [35], REL [44] and KBPearl [24]) for comparison, and these tools are introduced in Appendix D.3.

In addition, as introduced in Section 1, OKB entity linking is similar to the KB entity alignment task, but the 1-to-1 mapping assumption of KB entity alignment does not hold true for the OKB entity linking task. To comprehensively assess the effectiveness of our framework in OKB entity linking, we choose six advanced KB entity alignment methods which are not based on the 1-to-1 mapping assumption, for comparison, including: IPTransE [58], GCN-Align [50], MultiKE [56], BERT-INT [43], Dual-AMN [29] and RoadEA [39], whose setting details are shown in Appendix D.3.1.

**Table 3: Performance on OKB entity linking task.**

| Method | ReVerb45K | OPIEC59K |
|---|---|---|
| *Entity Linking Methods* | | |
| Tagme [13] | 0.316 | 0.662 |
| Spotlight [31] | 0.716 | 0.708 |
| Falcon [35] | 0.541 | 0.386 |
| REL [44] | 0.645 | 0.383 |
| KBPearl [24] | 0.522 | 0.552 |
| JOCL [27] | 0.761 | 0.757 |
| *KB Entity Alignment Methods* | | |
| IPTransE [58] | 0.519 | 0.392 |
| GCN-Align [50] | 0.672 | 0.810 |
| MultiKE [56] | 0.527 | 0.359 |
| BERT-INT [43] | 0.576 | 0.626 |
| Dual-AMN [29] | 0.710 | 0.805 |
| RoadEA [39] | 0.637 | 0.740 |
| CLUE | **0.864** | **0.876** |

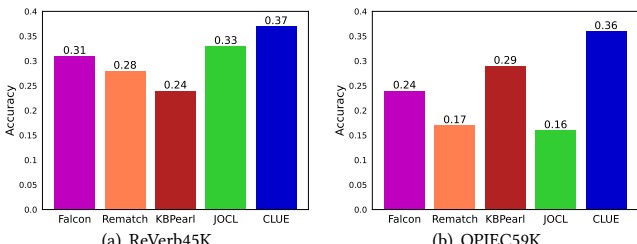

(a) ReVerb45K      (b) OPIEC59K

**Figure 2: Performance on OKB relation linking task.**

We show the experimental results for OKB entity linking in Table 3 where the accuracy performance of each baseline is obtained via running its open-source solution or using its publicly available API. It can be seen that all the entity linking methods, including JOCL that is specifically designed for OKB linking, perform poorly, which verifies that the task of OKB entity linking is non-trivial. Moreover, all the six KB entity alignment methods exhibit unsatisfactory performance, which may be attributed to the fact that these methods fail to leverage reciprocal benefits from the OKB canonicalization task. Overall, CLUE consistently exceeds all baselines over the two data sets by a large margin, which demonstrates the effectiveness of our framework for the task of OKB entity linking.

Combining the experimental results of OKB NP canonicalization and OKB entity linking, we could gain deeper insight by comparing our proposed CLUE with JOCL. Both CLUE and JOCL are designed to handle OKB canonicalization and OKB linking tasks jointly. However, CLUE still surpasses JOCL significantly on both tasks over two data sets, which implies that CLUE enables better and deeper interaction between OKB canonicalization and OKB linking with the help of the unified embedding space.

*4.3.2 OKB Relation Linking.* Apart from the aforementioned Falcon [35], KBPearl [24] and JOCL [27], we add Rematch [32] as a baseline for the task of OKB relation linking. Since RPs are not annotated, we randomly sample 100 OKB triples of both data sets and manually label each RP as the ground truth, the same as the previous work JOCL [27]. The experimental results for OKB relation linking are shown in Figure 2. To be specific, we obtain the results of all the baselines by executing the open-source solutions [24, 27] or leveraging the publicly available APIs [32, 35]. As shown in Figure 2, although the performances of the baselines over the two data sets differ a lot, our proposed CLUE consistently outperforms

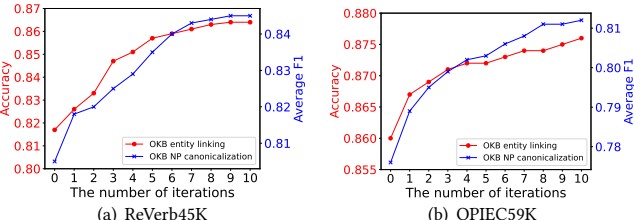

(a) ReVerb45K     (b) OPIEC59K

**Figure 3: Performance of CLUE as EM iterations progress.**

all the four baselines on both data sets, verifying the effectiveness of CLUE in OKB relation linking.

### 4.4 Effect Analysis of Iterative EM Process

To investigate the effectiveness of the iterative EM process (Section 3.3 and Section 3.4), we show how the canonicalization average F1 and linking accuracy achieved by CLUE change with respect to the number of iterations in Figure 3. We could see that the performance of CLUE on both OKB canonicalization and OKB linking tasks increase monotonically as EM iterations progress. Ultimately, the whole framework CLUE (w.r.t. the result at iteration 10) promotes by 4 (resp. 3.6) in terms of canonicalization average F1 and 4.7 (resp. 1.6) percentages in terms of linking accuracy over ReVerb45k (resp. OPIEC59K), compared with a variant of CLUE without the iterative EM process (w.r.t. the result at iteration 0). This confirms that the proposed EM based approach indeed effectively enhances the quality of the unified embedding space in an iterative manner by exploiting the deep coupling of OKB canonicalization and OKB linking. In addition, it is observed that the increasing speed of the average F1 and accuracy slows down as the number of iterations increases, demonstrating that convergence is achieved within the pre-defined ten iterations.

### 4.5 Ablation Study

To examine the effectiveness of different parts in the E-step of our framework CLUE, we conduct an ablation study by considering the following variants: (1) CLUE-w/o-CSG in which canonicalization seed generation (Section 3.3.1) is removed; (2) CLUE-w/o-LSG in which linking seed generation (Section 3.3.2) is removed; and (3) CLUE-w/o-CL that removes the curriculum learning based data selection (Section 3.3.3) but instead sets a fixed threshold for identifying high-confidence linking pairs and high-quality canonicalization clusters. We present the performance of these three variants as well as the whole framework CLUE on ReVerb45K in Figure 4. From the experimental results, we can see that CLUE outperforms CLUE-w/o-CSG and CLUE-w/o-LSG on both tasks, which validates that both canonicalization seeds and linking seeds generated in the E-step are beneficial for the refinement of the unified embedding space and further promote the performance of both tasks. Besides, compared with CLUE-w/o-CL, CLUE promotes by 1.5 (resp. 1.2) percentage in terms of canonicalization average F1 (resp. linking accuracy), indicating that the curriculum learning based data selection strategy could indeed adaptively recognize high-confidence linking pairs and high-quality canonicalization clusters, and achieve better performance than the fixed threshold method.

### 5 RELATED WORK

For the task of OKB canonicalization, previous methods can be classified into two types: (i) signal based methods which obtain

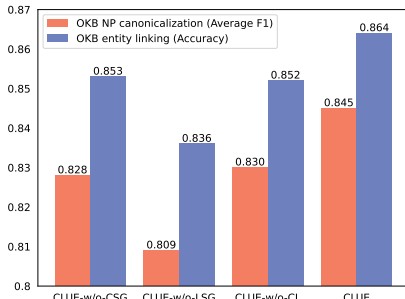

**Figure 4: Performance of different variants of CLUE.**

the canonicalization result according to diverse well-designed signals; and (ii) embedding based methods which perform clustering over the learned embeddings for NPs and RPs. Specifically, the first work for OKB canonicalization [14] belongs to the former type and uses several manually-defined signals, such as word overlap, IDF token overlap and Horn rules to get equivalent NPs and RPs. To improve the performance, SIST [23] obtains more signals by incorporating side information from the source text (i.e., candidate entities of NPs, entity typing information and domain knowledge). As for the embedding based methods, their performance highly depends on the quality of the learned embeddings. CESI [45] leverages side information as constraints for the loss function to learn better embeddings, which result in better overall canonicalization performance. CUVA [45] uses variational deep autoencoders to jointly learn both embeddings and cluster assignments in an end-to-end way. CMVC [37] learns view-specific embeddings for two views (i.e., fact view and context view) respectively and integrates the complementary knowledge from these two views via a multi-view K-Means clustering algorithm.

Nevertheless, all aforementioned works only handle OKB canonicalization in isolation, omitting the reciprocal benefits from the OKB linking task, which is inherently complementary with OKB canonicalization. JOCL [27] is the first and only work so far to handle OKB canonicalization and OKB linking tasks jointly and make them reinforce each other. By feeding the signals of OKB canonicalization (e.g., word embedding, PPDB [34], AMIE [15] and Stanford KBP system [42]) and the signals of OKB linking (e.g., entity popularity and Levenshtein distance) to the constructed factor graph, JOCL can solve these two tasks separately. Then, consistency signals are added to the factor graph to mutually constrain the output results of OKB canonicalization and OKB linking according to the coupling of these two tasks.

### 6 CONCLUSION

In this paper, we propose a novel framework CLUE which can resolve OKB canonicalization and OKB linking simultaneously and make these two tasks mutually reinforce each other via a shared and unified embedding space for encoding the OKB and CKB jointly. In order to refine the unified embedding space and further improve the performance of both tasks, an EM based approach is developed to perform seed generation and embedding refinement alternately, by utilizing the deep coupling of OKB canonicalization and OKB linking. Extensive experiments over two public benchmark data sets show that CLUE surpasses all the baselines for both OKB canonicalization and OKB linking tasks.

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

# A  PSEUDO-CODE OF FRAMEWORK CLUE

---
**Algorithm 1 CLUE**

---
**Input:** the sets of triples in OKB $T_o^+$ and triples in CKB $T_c^+$, the sets of original canonicalization seeds $\mathcal{V}_0^+$ and linking seeds $\mathcal{S}_0^+$, the number of iterations $K$

  Apply $T_o^+, T_c^+, \mathcal{V}_0^+, \mathcal{S}_0^+$ to learn the initial unified embedding space $\Theta_0$ by Eq. 1

  **for** $k = 1$ to $K$ **do**

    **E-step**:

      *1) Canonicalization Seed Generation:*

      Perform OKB linking based on the unified embedding space $\Theta_{k-1}$

      Calculate the confidence score of each linking pair by Eq. 7

      Obtain high-confidence pairs via data selection strategy in Section 3.3.3

      Generate the set of updated training canonicalization seeds $\mathcal{V}_k^+$ by Eq. 8

      *2) Linking Seed Generation:*

      Perform OKB canonicalization based on the unified embedding space $\Theta_{k-1}$

      Calculate the quality score of each canonicalization cluster by Eq. 9

      Obtain high-quality clusters via data selection strategy in Section 3.3.3

      Generate the set of updated training linking seeds $\mathcal{S}_k^+$ by Eq. 12

    **M-step**:

      Apply $T_o^+, T_c^+, \mathcal{V}_k^+, \mathcal{S}_k^+$ to learn the refined embedding space $\Theta_k$ by Eq. 20

  **end for**

  Perform OKB canonicalization and OKB linking based on the ultimately refined unified embedding space $\Theta_K$

**Output:** the final results of OKB canonicalization and OKB linking

---

# B  DESCRIPTIONS OF EVALUATION METRICS

## B.1  Metrics of OKB Canonicalization

As used in previous works [7, 14, 23, 27, 37, 45], we utilize macro, micro, and pairwise metrics to evaluate the OKB canonicalization result from different perspectives:

- Macro evaluates whether the NPs (or RPs) with the same semantic meaning have been clustered.
- Micro evaluates the purity of the resulting clusters.
- Pairwise evaluates individual pairwise merging decisions.

For each of these metrics, F1 score is defined as the harmonic mean of precision and recall. To give an overall evaluation of each OKB canonicalization method, we calculate average F1, which is a commonly recognized comprehensive metric, by averaging macro F1, micro F1, and pairwise F1.

## B.2  Metrics of OKB Linking

Following studies [27, 36], we adopt accuracy as the evaluation metric of OKB linking, calculated as the number of correctly linked NPs (resp. RPs) divided by the total number of NPs (resp. RPs).

# C  IMPLEMENTATION DETAILS

In our multi-task unified embedding learning model, we set the balance factors $\alpha_1$ and $\alpha_2$ both to 0.5. For the margin hyperparameters, we set $\gamma_1$ to 12, $\gamma_2$ to 12, and $\gamma_3$ to 20. The dimension of the unified embedding space is set to 300 and all the embeddings are initialized via fastText [17] trained on Common Crawl[2] . The learning rate is set to 0.0001 and the number of EM iterations $K$ is set to 10. When performing HAC for OKB canonicalization as described in Section 3.3.2, the Davies-Bouldin index [8] is used to predict the number of clusters.

---
[2]https://commoncrawl.org/2017/06

# D  DESCRIPTIONS OF BASELINE METHODS

## D.1  OKB NP Canonicalization Baselines

- Morph Norm [11] groups identical NPs after applying some simple normalization operations (e.g., removing tenses and pluralization).
- Text Similarity [14] clusters NPs by performing HAC using the Jaro-Winkler similarity [51] between them.
- IDF Token Overlap [14] utilizes HAC for NP clustering by calculating the inverse document frequency (IDF) token overlap as the similarity between NPs.
- Attribute Overlap [14] performs HAC for NP canonicalization based on the Jaccard similarity of attributes between two NPs.
- CESI [45] clusters the embeddings of NPs and RPs learned from the OKB triples and side information via HAC.
- SIST [23] performs OKB canonicalization by incorporating side information involved in the source text.
- CUVA [7] jointly learns embeddings and cluster assignments via variational autoencoders.
- JOCL [27] is the first framework for joint OKB canonicalization and linking, which is based on the factor graph model and leverages diverse signals like word embedding and PPDB [34].
- CMVC [37] is the state-of-the-art method for OKB canonicalization by leveraging two views of knowledge (i.e., fact view and context view) together.

## D.2  OKB RP Canonicalization Baselines

- AMIE [15] judges whether two RPs should be grouped by learning Horn rules.
- PATTY [33] can group OKB triples with the same pairs of NPs as well as RPs belonging to the same synset in PATTY.

## D.3  OKB Entity Linking Baselines

- Tagme [13] is a commonly used baseline for entity linking of short text fragments.
- Spotlight [31] is a well known entity linking baseline based on DBpedia [21].
- Falcon [35] conducts joint entity linking and relation linking via several basic principles of English morphology.
- REL [44] is a recent open-source entity linking toolkit, building on state-of-the-art neural components from NLP research.
- KBPearl [24] utilizes the facts and the side information from the context to jointly link NPs and RPs.

*D.3.1  Setting details of KB entity alignment baselines.* Since some KB entity alignment methods [39, 43, 56] were proposed to exploit attribute information, which is not available in OKBs, we have to remove the corresponding modules of such methods when applying them to OKB entity linking. Moreover, to facilitate a fair comparison across methods, we follow [22] to homogenize the matching module of KB entity alignment methods by searching the same dictionary as CLUE and provide them with an identical set of original linking seeds as ours for input.

