# OpenReview forum: "Jointly Canonicalizing and Linking Open Knowledge Base via Unified Embedding Learning"
_ACM.org/TheWebConf/2024/Conference — TheWebConf24 Oral_

### Official Review · Reviewer_hRZS · 2023-11-22

**Novelty:** 6
**Technical Quality:** 6

**Review:**

This paper focuses on the integration between curated knowledge bases (CKBs) on the Web, such as DBpedia, Wikidata, among others, and open knowledge bases (OKBs) like ReVerb or OPIEC. To that end, they divide the integration task in two subproblems, namely OKB canonicalization (grouping semantically similar terms) and OKN linking (discovering correspondences between terms in a OKB and their associated entities in a, CKN). The authors propose CLUE, a novel framework to address both subproblems in an unified manner. A number of experiments show the value of this approach.

The paper describes an interesting work with a lot of potential, since the method could be applied not only to already built OKBs but to any text corpus after applying triple extraction. The document is well written, well structured, and it is easy to follow. The core idea of the described framework it to build a common embeddings space to encode both the OKB and the CKB, later refined through an expectation-maximization algorithm. This is a really interesting approach, and its validity seems to have been confirmed by the reported experiments. I have no objections to this work, besides a few minor remarks:

A cite to the "expectation maximization" algorithm should be added on the first time it appears in the paper (in the Introduction). The same for "curriculum learning"

Figure 1 needs a better title to be more self-contained and not make it necessary to go back to the running text to understand it.

The relation of this approach with the web technologies should be made more explicit in the introduction, to better fulfil the relevance criteria of the Web Conference

UPDATE: I acknowledge having read the rebuttal and thank the authors for the clarifications

**Questions:**

no questions

**Reviewer Confidence:**

3: The reviewer is confident but not certain that the evaluation is correct

**Scope:**

4: The work is relevant to the Web and to the track, and is of broad interest to the community

---

### Official Review · Reviewer_woW4 · 2023-11-24

**Novelty:** 5
**Technical Quality:** 5

**Review:**

This paper designs an expectation-maximization based approach to iteratively refine the unified embedding space via performing seed generation and embedding refinement alternately, by leveraging the deep interaction between Open Knowledge Bases (OKB)canonicalization and OKB linking. Curriculum learning is employed to yield high-quality canonicalization seeds and linking seeds adaptively, according to two elaborately designed metrics (i.e., a margin-based linking metric and an entropy-based cluster metric).

The main pros:
1. This paper is the first to jointly encode the OKB and Curated Knowledge Base (CKB)into a shared and unified embedding space, to tackle OKB canonicalization and OKB linking simultaneously and make them reinforce each other.
2. Comprehensive experiments are conducted to demonstrate the effectiveness of the proposed method.
3. This paper is well organized and written.

The main cons:
1. Lack of state-of-the-art baselines for comparison.
2. Lack of deep experimental analyses. For example, Table 2 reveals that the proposed method does not consistently outperform the baselines in terms of macro F1, micro F1, and pairwise F1. What are the underlying reasons behind?
3. There are some presentation problems. For example, “The overall architecture of framework CLUE” should be “The overall architecture of CLUE”.

**Questions:**

Does the proposed method perform better than LLMs, such as ChatGPT?

**Ethics Review Description:**

None.

**Reviewer Confidence:**

4: The reviewer is certain that the evaluation is correct and very familiar with the relevant literature

**Scope:**

4: The work is relevant to the Web and to the track, and is of broad interest to the community

---

### Official Review · Reviewer_9hM3 · 2023-11-28

**Novelty:** 4
**Technical Quality:** 4

**Review:**

This paper studies a joint method for open knowledge base canonicalization and linking. It embeds OKB and CKB in a shared space, and proposes an EM algorithm to refine the space by finding high confidence seeds. Experiments show the proposed method can outperform baselines on canonicalization and linking tasks.

The paper is clearly written with technical details. The idea of joint embedding and iterative refinement with EM algorithm also makes sense for the joint modeling task. Experiments are conducted on two datasets and a range of baselines, showing the effectiveness of the proposed method.

One concern is that the proposed method seems to stack lots of components while the ablation studies are limited for showing the contribution of each claimed novelty. The paper also lacks some description on model efficiency analysis which can be important for applicability of an iterative method. Finally, there are some tables showing the proposed method can only achieve marginally improved performance, so some more explanation could be helpful for better understanding the paper.

**Questions:**

See above

**Reviewer Confidence:**

3: The reviewer is confident but not certain that the evaluation is correct

**Scope:**

3: The work is somewhat relevant to the Web and to the track, and is of narrow interest to a sub-community

---

### Official Review · Reviewer_uiBP · 2023-11-28

**Novelty:** 5
**Technical Quality:** 6

**Review:**

This paper presents a framework called CLUE for jointly canonicalizing and linking open knowledge bases (OKBs) by jointly encoding open knowledge bases (OKBs) and curated knowledge bases (CKBs) into a unified embedding space and performing OKB canonicalisation and OKB linking simultaneously reinforcing each other.


Pros:
- the proposed novel approach addresses the issue of leveraging the joint information while resolving the two related tasks, canonicalisation and linking, simultaneously using expectation-maximization and curriculum learning using two well-defined metrics.
- the proposed approach performs better than the state-of-the-art approaches in two relevant datasets (ReVerb45K, OPIEC59K). The proposed approach is evaluated on OKB NP canonicalisation, RP canonicalisation, entity linking tasks, and relevant baselines for each task are used for comparison. An ablation study is provided to show the contribution of different components of the approach.
- The approach is well-defined and sound, it is described in the paper with details and the source code is provided in an anonymous Git repo.

Cons:
- The paper lacks a discussion on the complexity and training/inference costs of the approach for users to have an idea of the cost/benefit of using this novel approach compared to the existing state-of-the-art approaches.

**Questions:**

- For most of the NLP tasks, the current state-of-the-art is significantly improved by applying large language models and foundation models. I believe such LLMs combined with approaches such as Retrieval Augmented Generation (RAG) are capable of performing tasks such as canonicalization and even entity reconciliation (e.g, https://arxiv.org/abs/2310.11244, https://arxiv.org/abs/2307.00524).

**Reviewer Confidence:**

3: The reviewer is confident but not certain that the evaluation is correct

**Scope:**

3: The work is somewhat relevant to the Web and to the track, and is of narrow interest to a sub-community

---

### Decision · Program_Chairs · 2024-01-22

**Decision:**

Accept (Oral)

**Comment:**

The paper suggests and approach to canonicalizing and linking open knowledge bases, by using a unified embedding space. The reviewers agree that the approach is novel and interesting, as well as relevant to the conference scope and track, although the latter could be better emphasised in the introduction. The paper is overall well written and clear, but lacks some discussions on the results, e.g. error analyses and analyses of the cases where improvement over the state of the art is not achieved, as well as analysis of the complexity and costs of the method etc.These things could be improved in the final version of the paper.